# Evaluation of the Physical Characteristics and Chemical Properties of Black Soldier Fly (*Hermetia illucens*) Larvae as a Potential Protein Source for Poultry Feed

**DOI:** 10.3390/ani13142244

**Published:** 2023-07-08

**Authors:** Rattanakorn Pornsuwan, Padsakorn Pootthachaya, Pasakorn Bunchalee, Yupa Hanboonsong, Anusorn Cherdthong, Bundit Tengjaroenkul, Wuttigrai Boonkum, Sawitree Wongtangtintharn

**Affiliations:** 1Department of Entomology and Plant Pathology, Faculty of Agriculture, Khon Kaen University, Khon Kaen 40002, Thailand; rattanakornp@kkumail.com (R.P.); yupa_han@kku.ac.th (Y.H.); 2Department of Animal Science, Faculty of Agriculture, Khon Kaen University, Khon Kaen 40002, Thailand; padsakornp@kkumail.com (P.P.); anusornc@kku.ac.th (A.C.); wuttbo@kku.ac.th (W.B.); 3Department of Biology, Faculty of Science, Mahasarakham University, Mahasarakham 44150, Thailand; pasakorn.b@msu.ac.th; 4Department of Veterinary Public Health, Faculty of Veterinary Medicine, Khon Kaen University, Khon Kaen 40002, Thailand; btengjar@kku.ac.th; 5Network Center for Animal Breeding and Omics Research, Faculty of Agriculture, Khon Kaen University, Khon Kaen 40002, Thailand

**Keywords:** black solider fly, protein insect meal, physico-chemical properties, feedstuff

## Abstract

**Simple Summary:**

There is a growing awareness of and increasing demand for products of animal origin that are sustainably produced in response to environmental concerns within the animal agriculture industry. This awareness has led to significant demand for alternative protein sources that are less reliant on conventional feed ingredients. In this context, insect meals represent a novel type of feed material that has been suggested as one of the possible alternatives that might solve this problem. Black soldier fly larvae (BSFL) are considered a sustainable protein source that is rich in nutrients. They are produced in an environmentally sustainable way that reduces the environmental impact associated with traditional protein sources. Therefore, the aim of this study was to evaluate the physical and chemical properties of BSFL with and without fat by using different drying methods. The results show that the various types of BSFL, drying methods, and their interactions influenced the physico-chemical composition of the feed. The presented data show that BSFL could be considered for use in poultry feed instead of conventional protein.

**Abstract:**

The aim of this study was to investigate and compare the effects of different drying methods on the physical and chemical properties of black soldier fly larvae (BSFL) to determine their potential as an alternative protein source in animal feed. The experimental design was a 2 × 3 factorial arrangement in a completely randomized design (BSFL type × drying method), with five replications. The influence of post-harvest procedures was studied, including the different BSFL types (non-defatted and defatted) and drying methods (parabola dome, hot air oven, and microwave). The results showed that the types of BSFL, drying methods, and their interaction significantly (*p* < 0.001) influenced the feed’s physical properties; these included the brightness of color (*L** 29.74–54.07; *a** 0.40–5.95; *b** 9.04–25.57), medium bulk density (381.54–494.58 g/L), free flow with an angle of repose (41.30–45.40°), and small particle size. They significantly (*p* < 0.001) influenced the nutritive value of BSFL, which contained 42–59% crude protein, 7–14% crude fiber, 9–30% ether extract, and 5035–5861 kcal/kg of energy. Overall, both BSFL types and all the drying methods resulted in a slight variation in the proximate composition. However, a microwave and a hot-air oven were considered the most suitable methods for producing BSFL powder because of the high levels of nutrients retained and the improved physical parameters when compared to a parabola dome. This characterization of the physical and chemical composition of BSFL represents a preliminary methodology that could be used to initially preprocess larvae for use as an alternative protein source in animal feed and for other applications.

## 1. Introduction

Black soldier fly larvae (BSFL) have received widespread attention as an alternative source of animal feedstuffs due to their nutritional benefits, environmental sustainability, and potential for economic efficiency. BSFL are highly nutritious, have a balanced amino acid profile, and feature high protein, fat, and mineral levels [1,2]. Therefore, they could serve as an excellent source of sustainable protein and other essential nutrients, making them an ideal ingredient for aquaculture feed [3]. In addition to aquaculture, there is growing interest in using BSFL in livestock and poultry feed [1,4], as well as in food for companion animals, which is gaining great popularity. However, in many countries, insect meal is still too expensive for introduction into livestock feed.

The black soldier fly (*Hermetia illucens*) can efficiently convert organic waste materials (i.e., food scraps, agricultural by-products, and manure) into protein-rich larvae [5]. This can help to reduce the environmental impact of these waste materials and convert them into valuable protein sources for animal feed. Their use could promote a circular economy approach by reducing waste, conserving resources, and minimizing the use of traditional feed ingredients, such as fishmeal and soybean meal, which may have negative environmental impacts associated with their production and sourcing. In addition, when comparing the economics of commercial BSFL use, insect farming has lower production costs than soybean meal production and has a faster growth rate, allowing for quicker turnover and a potentially higher yield per unit of the resources invested. However, the use of manure or food scraps may be unacceptable in several countries due to veterinary and legal regulations. Insects must only consume approved products in order to enter the food chain.

The growing awareness of sustainability and the rise in environmental concerns among consumers and the animal agriculture industry has resulted in a demand for alternative production systems that are less reliant on conventional feed ingredients. BSFL represent an interesting option and are considered a sustainable protein source. The inclusion of BSFL in animal feed has been shown to improve animal growth, performance, and feed efficiency [1,6]. As the use of BSFL in animal feed gains popularity, there has been an increasing focus on constructing regulatory frameworks and standards to ensure safety, quality, and sustainability. Various organizations and regulatory bodies, such as the European Food Safety Authority (EFSA) and the U.S. Food and Drug Administration (FDA), have provided guidance and regulations for using BSFL in animal feed [7,8].

Recently, the utilization of BSFL as a novel protein source and its nutritional composition have been extensively investigated by researchers. For instance, Spranghers et al. [9] conducted a study to examine the nutritional components of BSFL prepupae that were reared on various organic waste substrates. The results revealed only minor differences in the amino acid profiles of the prepupae. Additionally, the impact of technical processes such as extrusion [10], defatting [11], and extraction [12] on the nutritional value of BSFL has been documented. However, the drying process plays a critical role in determining the nutritional value of commercial BSFL. Among the various drying techniques used in the food processing industry, microwave radiation has gained popularity compared to ovens or solar dryers. There are more than 20 countries that already offer irradiated meat products commercially around the world [13]. Despite this, there are few studies examining the influence of drying methods on the nutritional value of BSFL.

The physical characteristics of alternative feedstuffs are important when planning feed rations. They impact the decisions that must be made when planning and designing on-site farm feed storage. Discussing the basic terminology and physical properties of alternative feeds should help with the recognition of essential considerations, such as the volume of storage required and the handling options that need to be addressed in the planning stages [14]. The chemical composition of animal feed plays a crucial role in meeting the nutritional requirements of animals, improving feed efficiency, maintaining animal health, ensuring product quality, and minimizing environmental impacts [14,15]. It is essential for farmers, nutritionists, and animal producers to carefully formulate and manage animal feed to optimize animal performance and welfare, as well as to achieve sustainable and profitable animal production.

Therefore, this study aimed to evaluate the influences of drying methods, i.e., the use of the parabola dome, hot-air oven, and microwave drying methods, on the physico-chemical properties of BSFL with and without fat and its potential as an alternative protein source for use in animal feed in Thailand.

## 2. Materials and Methods

### 2.1. Black Soldier Fly Larvae (BSFL) Preparation

The BSFL were reared at the Model House for Research and Production of Industrial Insects, Industrial Insect Division, Faculty of Agriculture, Khon Kaen University. The BSFL were fed with mixed feed, consisting of 22.5% peanut seed coats, 22.5% fruit waste, 22.5% slaughterhouse waste, 22.5% food scraps, and 10.0% water, with nutritional values consisting of approximately 50% DM and 15% CP. Slaughterhouse and fruit wastes were ground down to allow passage through a 2.0 mm screen. Then, they were mixed with the remaining feed and offered *ad libitum* to the BSFL. In the process of BSFL preparation, samples of about 36 kg of BSFL in the prepupae period, aged 26–28 days, were rinsed thoroughly under running tap water for 1 min to wash away the contaminants, as well as to wash the larvae. The living insects were sacrificed by freezing at −20 °C for 24 h in the refrigerator and used to investigate the effect of drying by different methods and types of BSFL, both full-fat and defatted. The BSFL were divided into three groups of 12 kg each and were prepared using different drying methods, comprising a parabola dome, hot-air oven, and microwave.

(I)Parabola dome drying procedure: First, 12 kg of BSFL were spread evenly in an aluminum tray, and they were then dried inside the parabola dome for 48 h until a moisture content of not more than 10% was achieved.(II)Hot-air oven procedure: The oven (30-UF1060, Memmert GmbH + Co. KG, Schwabach, Germany) was preheated for 15 min until it reached the required temperature. Afterward, 12 kg of BSFL were spread evenly in an aluminum tray and were then dried at 60 °C for 48 h until a moisture content of not more than 10% was achieved.(III)Microwave drying procedure: A sample of 1 kg of BSFL was placed in a plastic bucket (1 kg per bucket for a total of 12 times) and was dried until a moisture content of not more than 10% was achieved. This occurred inside a microwave (Microwave Vacuum Rotary Dryer, March Cool Industry Co., Ltd., Bangkok, Thailand) under vacuum conditions at a power level of 800 watts for 10 min, during which time the baking temperature did not exceed 60 °C.

Samples of dried BSFL obtained using each drying method were divided into two parts containing the non-defatted and defatted BSFL by a physical process, resulting in a total of 6 experimental groups, in various combinations: non-defatted BSFL dried with a parabola dome (NDP-BSFL); non-defatted BSFL dried with a hot-air oven (NDH-BSFL); non-defatted BSFL dried with a microwave (NDM-BSFL); defatted BSFL dried with a parabola dome (DP-BSFL); defatted BSFL dried with a hot-air oven (DH-BSFL); defatted BSFL dried with a microwave (DM-BSFL). To ascertain the physical characteristics and for chemical composition analysis, a representative of the BSFL from each group was ground with a mill grinder (MF 10 basic microfine grinder drive, IKA^®^-Werke GmbH & CO. KG, Staufen, Germany) until fine enough to allow passage through a 1.0 mm sieve screen for the purposes of laboratory analysis.

### 2.2. Physical Characteristic Measurement

#### 2.2.1. Color

A sample of 5 g of BSFL was also taken from each group and used for color measurement. The color of the samples was determined using the CIELAB system (*L**, *a**, and *b**), along with the Chroma Meter (CR-410 Series, Konica Minolta Sensing Inc., Tokyo, Japan). The CIELAB color space system consists of the *L** value, which measures lightness on a scale from 0 (black) to 100 (white), the *a** axis, which measures redness (positive) to greenness (negative), and the *b** axis, which measures yellowness (positive) to blueness (negative). Each group was tested with 5 replications. Color variation (ΔE**_Lab_*) was calculated following Purschke et al. [16].
$${\Delta E*}_{\mathrm{Lab}}=\surd {\left(\Delta L*\right)}^{2}+{\left(\Delta a*\right)}^{2}+{\left(\Delta b*\right)}^{2}$$

#### 2.2.2. Bulk Density

The bulk density was measured following the method described by ASABE [17]. The sample was measured using a 1000 mL cylinder. The bulk density was calculated by dividing the weight of the sample by the volume of space that it occupies. Each sample measurement was replicated 5 times.

#### 2.2.3. Angle of Repose

Measurement of the angle of repose of the BSFL was performed by pouring 100 g of each group from a certain height through a conical funnel onto a flat surface. The funnel had an outer diameter of 1 cm and was fixed using a metal stand. The outlet of the funnel was positioned 6 cm above the floor. The angle that the cone-shaped sample formed on both the right and left sides was then measured using a protractor, following the method described by Carr [18]. Each sample measurement was replicated 10 times.

#### 2.2.4. Particle Size and Distribution

The particle size distribution was measured using an electric sieve shaker, which consisted of six mesh sizes of 20 (850 μm), 40 (425 μm), 60 (250 μm), 80 (180 μm), and 100 (150 μm) mesh (Laboratory Test Sieve Series, Endecott’s Ltd., London, UK). The shaker was set at a 2.5 mm amplitude and was operated for 20 min with 100 g of BSFL from each group. Each sample measurement was replicated 5 times. After the shaking process, the weight of each layer was recorded to calculate the percentage of retention and determine the passing of particles. The geometric mean diameter or median size of the particles (*D_gw_*) was determined following the method described by Oryza et al. [15] as follows:Retain (%) = [Total sample weight in the sieve/Total weight of sample] × 100
Passing (%) = 100 − retain (%)
$$D_{\mathrm{gw}}=\log^{-1}\;[\sum_{i=1}^{n}(w_{ilogdi})\;/\;\sum_{i=1}^{n}w_{i}]$$
where *w_i_* is the mass in each sieve (g) and *d_i_* is the sieve size (mm), which is calculated as (*d* × *d*_+1_)^0.5^. The calculation of the geometric mean diameters or median size of the particle (*D_gw_*) follows the method described by Vasconcelos et al. [19].

#### 2.2.5. Microscopy Compound

The morphology of BSFL from each group was observed using a stereo microscope (JSZ5B, Novel Optics Co., Ltd., Beijing, China) at magnifications of 20×, 40×, and 60×. Scanning electron microscopy (SEM) was employed to visualize the ultrastructural morphology of BSFL, following the methods described by Tanpong et al. [14] and Oryza et al. [15]. Briefly, 3 g of the sample was affixed to a stub using double-sided carbon adhesive tape and sputter-coated with a thin layer of gold (40–50 nm) under high-vacuum conditions. The morphological characteristics of the BSFL samples were analyzed using an SEM electron microscope (JSM 6460 LV, JEOL Ltd., Tokyo, Japan) at magnifications of 25×, 50×, 100×, and 300×, with an accelerating voltage of 15 kV. The images were collected from a series of 3 replicates per sample.

### 2.3. Chemical Composition Measurement

The BSFL meal was measured for nutritional composition using proximate analysis following the methods of AOAC [20], to assess the level of moisture (Method 930.15), ash (Method 942.05), crude fiber (Method 978.10), and crude fat (Method 920.39). Crude protein (N × 6.25) was analyzed using a nitrogen analyzer (FP828, LECO Instruments Co., Ltd., Bangkok, Thailand), and the nitrogen-free extract was calculated. Then, the gross energy content was analyzed using adiabatic bomb calorimeters (AC500, LECO Corp., St. Joseph, MI, USA). Both phosphorus and calcium contents were measured according to AOAC [20]. Phosphorus content was determined using the photometric method (Method 964.06) via an atomic absorption spectrophotometer (T80+, PG Instrument Ltd., Lutterworth, England) while the calcium content was assayed using the titration method (Method 927.02). Finally, the pH of sample was measured with a pH meter (PHS-3C, Shanghai Puchun Measure Instrument Co., Ltd., Shanghai, China). The samples were tested using 5 replications for each sample.

### 2.4. Statistical Analysis

The data were checked for normality using the Shapiro–Wilk test and for homogeneity of variance using Levene’s test; any outlier data were eliminated before analysis. Data on the physical characteristics and chemical composition of the BSFL were analyzed using a completely randomized design with a 2 × 3 (BSFL types (A) × drying methods (B)) factorial treatment structure. Statistical analyses were performed using a two-way analysis of variance (ANOVA) and were produced using SAS version 9.1 software. The significance of the differences among the groups was determined using Tukey’s studentized range test at *p* < 0.05. The statistical model is as follows:Y_ijk_ = μ + α_i_ + β_j_ + αβ_ij_ + ε_ijk,_
where Y_ijk_ = observation, μ = overall mean, α_i_ = BSFL type effect (i = non-defatted and defatted), β_j_ = drying method effect (j = parabola dome, hot-air oven, and microwave), αβ_ij_ = BSFL types × drying methods, and ε_ijk_ = error.

## 3. Results

### 3.1. Physical Characteristics of BSFL

#### 3.1.1. Color

The physical characteristics and color space values of the BSFL are shown in Figure 1 and Table 1. The results demonstrate that the different types of BSFL (A), the various drying methods (B), and their interaction (A × B) significantly (*p* < 0.001) influenced the values of *L**, *a**, *b**, and ΔE. The findings show that the brightness, redness, and yellowness values of the NDP-BSFL group were substantially lower (*p* < 0.05) than in the other groups. Moreover, the ΔE was significantly higher (*p* < 0.05) in the NDP-BSFL group than in the other groups.

#### 3.1.2. Bulk Density

The bulk density is given in Table 1. The different types of BSFL (A), drying methods (B), and their interaction (A × B) significantly (*p* < 0.001) influenced the bulk density of the various samples. The results show that the bulk density value of defatted BSFL was 17.32% higher (*p <* 0.05) than that of non-defatted BSFL. The mean bulk density using the microwave drying method was 1.33% and 3.69% higher (*p <* 0.05) than that of the hot-air oven and parabola dome drying methods, respectively. Meanwhile, the interaction effect demonstrated that the mean bulk density was the highest (*p <* 0.05) in the DH-BSFL group (494.59 g/L).

#### 3.1.3. Angle of Repose

The drying methods (B) and their interaction (A × B) also influenced the angle of repose (*p* = 0.002 and *p* < 0.001, respectively; Table 1). The angles of repose of the products of the hot-air oven and microwave drying methods were 43.65° and 43.35°, respectively, which were higher (*p <* 0.05) than that obtained when using the parabola dome drying method. Meanwhile, the interaction effect showed that the angle of repose was the highest (*p <* 0.05) in the NDM-BSFL group (45.40°).

#### 3.1.4. Particle Size and Distribution

The particle size and distribution are shown in Table 2. The retention percentage corresponds to the weight of the sample, which is shown by the percentage of BSFL in each section after passing the ground sample through 20, 40, 60, 80, and 100 mesh and into a pan. We found that the types of BSFL (A), the various drying methods (B), and their interaction (A × B) significantly (*p* < 0.001) influenced particle size. The average particle size of the non-defatted BSFL was significantly higher (*p* < 0.05) than that of the defatted BSFL at 20 (23.49%) and 40 (66.30%) mesh. In contrast, the defatted BSFL showed a particle size higher (*p* < 0.05) than the non-defatted BSFL at 60 (41.27%), 80 (24.11%), and 100 (9.77%) mesh and in the pan (0.93%). The average particle sizes of sample prepared using the parabola dome, hot-air oven, and microwave methods showed high retention percentages at a 40 mesh size (425 μm) of 49.01, 44.97, and 34.97%, respectively. The geometric mean diameter (*D_gw_*) was significantly highest (*p <* 0.001) in the NDM-BSFL group (414.98 μm).

#### 3.1.5. Microscopic Characterization

A stereo microscope (Figure 2 and Figure 3) and scanning electron microscope (Figure 4 and Figure 5) were used to observe differences in the particle size and ultrastructure of BSFL flour. There were differences in particle size and distribution between BSFL samples pretreated in a solar dryer dome, a hot-air oven, and a microwave oven, with and without lipid extraction. Furthermore, dispersive particles with small diameters were found when using the hot-air oven drying method (Figure 4b and Figure 5b). However, rigid and compact particles with a larger diameter were observed for the microwave drying method (Figure 4c and Figure 5c), whereas the surface morphology of the solar dryer dome sample showed large and blocky particles. Moreover, some irregular, cracked, or shrunken particles could also be observed, which are shown in Figure 4a and Figure 5a.

### 3.2. Chemical Composition of BSFL

The results of chemical analysis of the BSFL are shown in Table 3. The types of BSFL (A), the drying methods (B), and their interaction (A × B) significantly (*p* < 0.001) influenced moisture, ash, crude protein, crude fiber, ether extract, nitrogen-free extract, phosphorus, and gross energy content, while B also affected calcium (*p* = 0.006), but the various combinations did not influence pH (*p* > 0.05). The moisture, lipid, phosphorus, and gross energy contents of the non-defatted BSFL group were 5.23%, 31.12%, 0.70% DM, and 5767.04 kcal/kg, respectively, which were significantly (*p* < 0.05) higher than in the defatted BSFL group. In contrast, the ash, crude protein, crude fiber, and nitrogen-free extract values of the defatted BSFL group were 6.66, 56.60, 12.18, and 8.69% DM, respectively, which were significantly (*p* < 0.05) higher than for the non-defatted BSFL group. Among the different drying methods, the parabola dome drying method showed the highest (*p* < 0.05) moisture level and ash content (6.97 and 6.23% DM). The hot-air oven drying method showed the highest (*p* < 0.05) crude protein, crude fiber, ether extract, calcium, phosphorus, and gross energy contents (51.15, 11.01, 22.40, 1.70, 0.70% DM, and 5483.04 kcal/kg). The microwave drying method showed similar calcium and phosphorus contents to those of the sample using the hot-air oven method, but it also showed the highest values for nitrogen-free extract and gross energy (10.54% DM and 5498.62 kcal/kg). However, the interaction effect showed that the DM-BSFL group showed the significantly highest (*p* < 0.05) value regarding the contents of crude protein and nitrogen-free extract and the lowest (*p* < 0.05) levels of moisture and lipid content.

## 4. Discussion

The present study aimed to confirm the effects of various drying techniques on the physical properties and nutritional values of both non-defatted and defatted BSFL powder. The physico-chemical properties of feedstuff have a great influence on feed formulation. These properties can impact the processing, storage stability, nutrient availability, and overall performance of the feed. The physical properties include color, shape, texture, bulk density, angle of repose, particle size, and smell, which are relevant to the production, processing, and handling of feed [21]. These parameters are commonly used to determine the maximum feed storage capacity and its relationship with animal intake and utilization. This includes the levels of essential nutrients such as protein, fiber, carbohydrates, vitamins, and minerals, which must be considered in animal feed formulation, as they are critical for animal growth, health, and performance [22]. Knowing the nutritional value of animal feed helps farmers and animal nutritionists formulate balanced and appropriate diets to meet the specific nutritional requirements of different animal species at the various production stages.

The color of BSFL was observed to be affected by processing, as indicated by the color difference (ΔE**_Lab_*). According to Adekunte et al. [23], perceivable color differences can be categorized as “very distinct” (ΔE**_Lab_* > 3), “distinct” (1.5 < ΔE**_Lab_* < 3), and “small difference” (ΔE**_Lab_* < 1.5). In this study, the ΔE**_Lab_* of the hot-air oven and microwave drying methods with and without lipid extraction was in the range of 3.54–5.65, indicating that there were perceptible color differences at a glance. In other words, the color changes could be perceived without close inspection. Meanwhile, the maximal color deviation (ΔE**_Lab_* = 28.10), as indicated by low brightness (*L**), redness (*a**), and yellowness (*b**), was observed for the non-defatted BSFL obtained using the parabola dome method (NDP-BSFL), which exhibited a darker color when compared to other groups. The high ΔE**_Lab_* value for the NDP-BSFL group can be attributed partly to the phenolic compounds released in the insect’s cuticle or integument. As a result, the resulting product could undergo oxidation, protein–polyphenol interaction, and reaction with phenoloxidase, which catalyzes the browning of insect meals [2,24]. Moreover, the initial color was best conserved in the case of defatted BSFL, particularly both DH-BSFL and DM-BSFL, due to the omission of thermal browning reactions in the dried larvae and the co-extraction of brown lipophilic melanoidins during the defatting process. Therefore, our results for several color parameters suggested that drying techniques using either the oven or microwave could be used to improve the color of fresh BSFL and promote the sensory nutrition of BSFL compared to techniques using the solar dryer, due to the inactivation of browning enzymes and the Maillard reaction [16,25]. Nonetheless, the reaction mechanism resulting in the BSFL meal’s dark brown color is poorly understood. The defatted BSFL groups demonstrated a significant increase in bulk density compared to the non-defatted BSFL. These results are inconsistent with those of Mshayisa et al. [2], who reported no significant differences in bulk density between the freeze-dried and defatted BSFL flours. The bulk density of BSFL obtained by different drying methods was 381.54–494.58 g/L. The bulk density of a feed ingredient can vary depending on factors such as particle size, moisture content, and processing methods [14]. The typical bulk density range for soybean meal is around 500–700 g/L [26]. Therefore, in comparison to soybean meal, the bulk density of BSFL appears to be relatively lower (fairly good). In addition, bulk density affects animal feed intake; low bulk density will fill less gut space and may restrict the feed intake of animals [27,28].

The angle of repose of the BSFL meals produced in this research was classified as offering fair to passable flow (41.30–45.40°). According to Basava Raj et al. [29], the degree of angle of repose can be classified as “excellent” (25–30°), “good” (31–35°), “fair” (36–40°), “passable” (41–45°), “poor” (46–55°), and “very poor” (56–65°). Oryza et al. [15] reported that the flowability of powder is impacted by the physical characteristics of the material and the particular processing circumstances present in the handling system. The density and particle size, the surface features of materials, and the water and fat contents of the feedstuff all affect the angle of repose [14]. Feedstuffs with larger particle sizes tend to have a lower angle of repose. Additionally, the average angle of repose is dependent on the moisture content of the material.

The BSFL meal produced in the current study had a small particle size that could pass through each grade of sieve; most samples contained a particle size of between 180 and 850 μm or 20–80 mesh size. Moreover, the geometric mean diameter (*D_gw_*) of all the BSFL groups showed that the resulting BSFL meal had a small particle size. The particle size and shape influence the flow and compaction properties of the meal. Furthermore, particle size impacts the utilization and production characteristics of the animals receiving the produced feed, including nutrient digestibility, feed intake, gut health, and overall performance. Generally, animals prefer finer particles over coarser ones, as they are more palatable and have a higher surface area for taste and odor perception [30,31]. Examples include corn meal and soybean meal, which present particle sizes of between 300 and 800 μm [32]. Finely ground feed may result in higher feed intake, which can positively impact animal performance, because the small particles increase the feed’s surface area, making it more accessible to digestive enzymes and improving nutrient absorption [33], particularly in young animals with a limited capacity for feed consumption. Vu et al. [34] reported that larger particles could decrease the feed intake of animals by restricting the surface area per unit and inhibiting the enzymatic digestion of nutrients.

Regardless of drying method, all the BSFL groups, particularly the NDP-BSFL group, showed a darker color under a stereo microscope, which supported the physical analysis results (Table 1). Under a scanning electron microscope, the physical characteristics of the samples, including damage to dried BSFL in terms of their color, shape, and structure, showed that prolonged exposure to microwave radiation and heat might cause damage to the integrity of the cuticle, which acts as the outer protective layer of BSFL [35]. Additionally, microwave drying can potentially cause the deformation or distortion of BSFL due to the application of heat and moisture removal [36], which can be observed as cracks, fractures, or changes in the ultra-surface structure of the cuticle. Thus, the physical properties of dried BSFL were highly dependent upon the details of the drying techniques (i.e., temperature, time, shrinkage, heat transfer, and sample consistency).

The quantification and elucidation of the proximate composition of both BSFL powders are necessary to evaluate the drying methods that most influence the nutritional quality of the BSFL meal, which could be considered as a form of protein and an energy source rich in both proteins and lipids. The low moisture content of both defatted and non-defatted BSFL powders suggests that the three drying techniques can be used to prolong the shelf life of BSFL flour. This will ensure their availability all year round. However, the low moisture content of the microwaved BSFL powder compared to oven-dried powder suggests that the microwave drying technique can be used to further improve the quality of BSFL powder. In addition to moisture content, water activity is another crucial factor in determining a product’s shelf life. Controlling water activity is essential for preserving the quality, safety, and shelf life of feed ingredients [37]. Therefore, understanding the relationship between water activity, moisture content, and product characteristics is crucial in determining the appropriate storage conditions, packaging, and formulation strategies to achieve the desired shelf life for a specific product.

In terms of chemical composition, the results regarding protein content in this study were similar to those reported by Mshayisa et al. [2] in their study on BSFL (44–68% of DM) and were higher compared to the values reported by Huang et al. [38], who reported the average protein concentration of BSFL to be 42% DM. The differences are probably related to the drying conditions employed and the diet fed to BSFL [39]. The slight variations observed may be due to differences in the final moisture content resulting from the various drying techniques [33,40], as well as the defatting process. Higher moisture content in a feedstuff can result in a lower concentration of nutrients (such as proteins, fats, and carbohydrates) when assessed on a dry matter basis. The positive impact of final moisture content was observed in the form of the high proportion of protein and carbohydrate content after using the microwave drying method for BSFL meal with (43.16 and 8.99% of DM, respectively) and without fat (59.14 and 12.09% of DM, respectively) when compared to other groups.

However, higher proportions of crude fiber in BSFL meal can be problematic when fed to monogastric animals, particularly poultry, in which only a tiny percentage of crude fiber is broken down in the gastrointestinal tract. In poultry species, which have relatively short digestive tracts and rapid digestion processes, the ability to break down and utilize crude fiber is particularly limited [41]. Therefore, feeding diets with higher proportions of crude fiber to poultry can result in reduced feed efficiency, decreased nutrient absorption, and potentially negative impacts on growth and performance.

The current study demonstrated that the various drying approaches resulted in a slight to moderate variation in the protein, fat, fiber, and carbohydrate contents of BSFL meal. This finding suggests that the choice of drying method can influence the physico-chemical properties of the resulting BSFL product. However, an evaluation of the suitability of BSFL for use in poultry feed formulation is not only made according to its physical and chemical composition but is also influenced by other factors such as amino acid and fatty acid contents and digestibility. This is because the duration of drying, dehydration, or the temperature level of the drying method can cause protein denaturation, oxidation, and a lack of digestibility, which may all be affected by the processing process [37,42]. This factor should be considered in the second step of determining the suitability of BSFL meal for use in poultry diets.

## 5. Conclusions

The present study investigated the physico-chemical properties of BSFL as an alternative source of protein in animal feed. This study showed that the type of BSFL meal significantly influences the nutritional value, i.e., protein, fat, and fiber content. The contrasting drying methods used resulted in slightly different physical characteristics, including the brightness of color, the medium of bulk density, passable flow, small particle size, and chemical composition. However, the physico-chemical properties of the dried BSFL were found to be highly dependent on the applied drying technique. Comparing the different drying techniques, the microwave and oven drying techniques were more effective than using a parabola dome. Moreover, when considering the interaction effect of the BSFL type and the drying method, based on the physico-chemical properties, the results show that the DM-BSFL treatment was more effective than the other treatments. Nevertheless, the data reported in this study could represent preliminary guidelines for considering and handling raw materials before use in a feed formulation.

## Figures and Tables

**Figure 1 animals-13-02244-f001:**
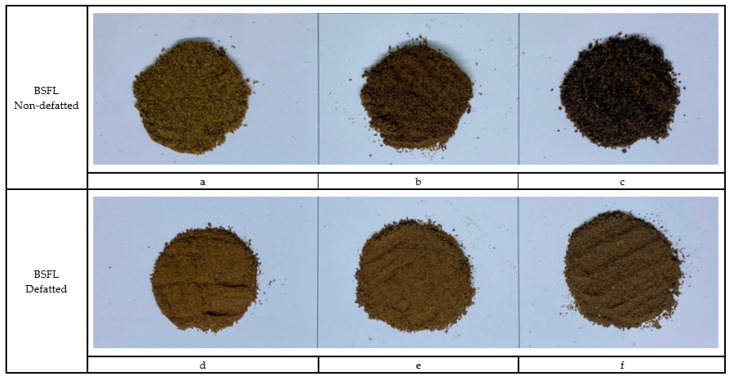
The physical characteristic of BSFL meal by types and drying methods. Non-defatted BSFL: (**a**) microwave; (**b**) hot-air oven; (**c**) parabola dome; defatted BSFL: (**d**) microwave; (**e**) hot-air oven; (**f**) parabola dome.

**Figure 2 animals-13-02244-f002:**
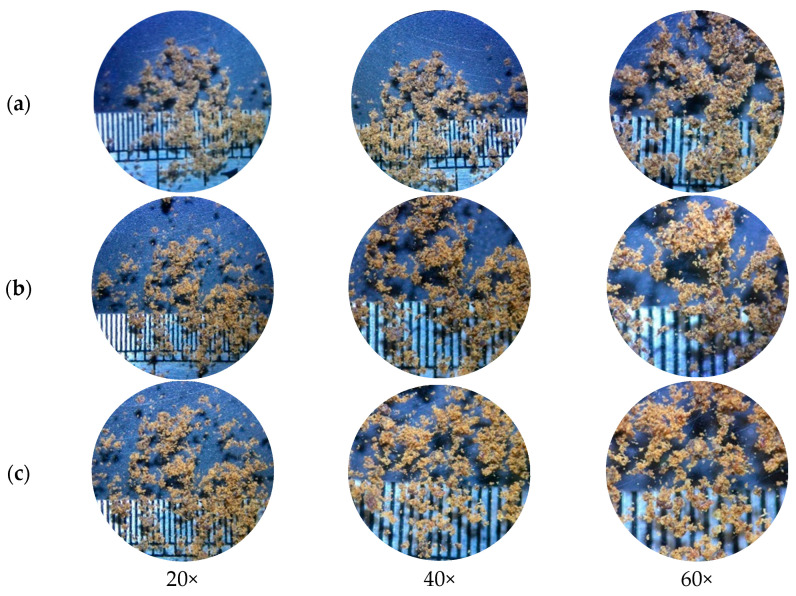
Stereoscopic micrographs of non-defatted BSFL. (**a**) Parabola dome, (**b**) hot-air oven, and (**c**) microwave at 20×, 40×, and 60× magnifications.

**Figure 3 animals-13-02244-f003:**
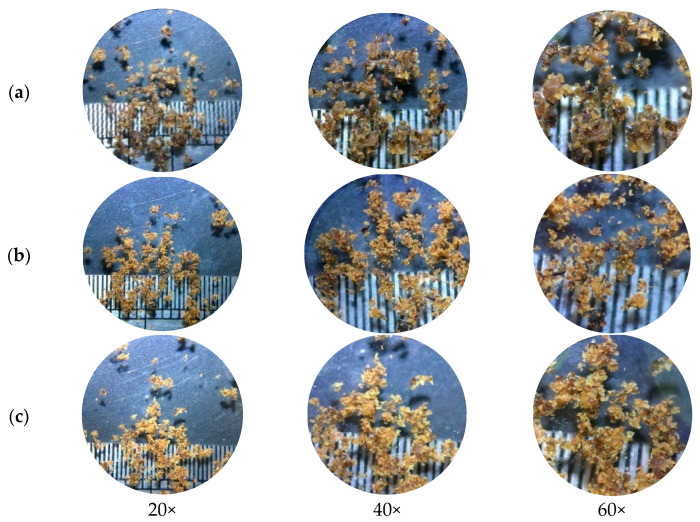
Stereoscopic micrographs of defatted BSFL. (**a**) Parabola dome, (**b**) hot-air oven, and (**c**) microwave at 20×, 40×, and 60× magnifications.

**Figure 4 animals-13-02244-f004:**
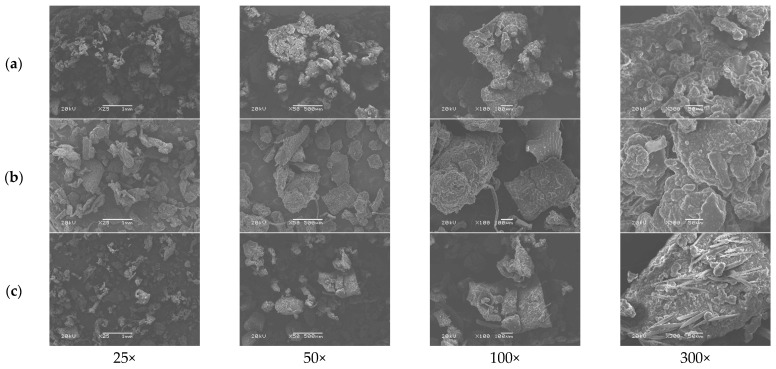
Scanning electron micrographs of non-defatted BSFL. (**a**) Parabola dome, (**b**) hot-air oven, and (**c**) microwave at 25×, 50×, 100×, and 300× magnifications.

**Figure 5 animals-13-02244-f005:**
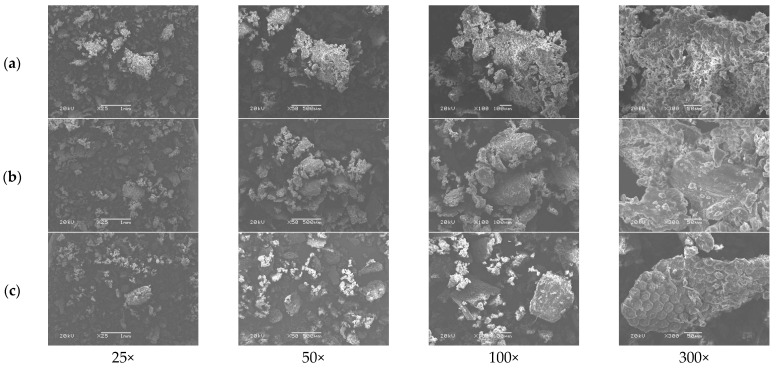
Scanning electron micrographs of defatted BSFL. (**a**) Parabola dome, (**b**) hot-air oven, and (**c**) microwave at 25×, 50×, 100×, and 300× magnifications.

**Table 1 animals-13-02244-t001:** Color parameter, bulk density, and angle of repose of BSFL after different drying methods.

Items	Color				Bulk Density	Angle of Repose
	*L**	*a**	*b**	ΔE ^1^	(g/L)	(°)
Raw material	54.96	4.74	20.64	-	-	-
Non-defatted BSFL						
Parabola dome (NDP)	29.74 ^b^	0.40 ^d^	9.04 ^c^	28.10 ^a^	385.64 ^d^	41.30 ^c^
Hot-air oven (NDH)	49.34 ^a^	4.38 ^ab^	20.24 ^b^	5.65 ^b^	381.54 ^d^	42.60 ^c^
Microwave (NDM)	54.07 ^a^	3.01 ^bc^	25.57 ^a^	5.30 ^b^	418.24 ^c^	45.40 ^a^
Defatted BSFL						
Parabola dome (DP)	52.54 ^a^	2.13 ^c^	19.54 ^b^	3.73 ^d^	469.52 ^b^	43.00 ^bc^
Hot-air oven (DH)	51.14 ^a^	4.28 ^b^	23.71 ^ab^	4.92 ^c^	494.58 ^a^	44.70 ^ab^
Microwave (DM)	52.54 ^a^	5.95 ^a^	22.92 ^ab^	3.54 ^e^	469.70 ^b^	41.30 ^c^
SEM	1.341	0.410	1.238	0.043	0.934	0.171
BSFL type means						
Non-defatted	44.38 ^b^	2.60 ^b^	18.28 ^b^	13.02 ^a^	395.14 ^b^	43.10
Defatted	52.07 ^a^	4.11 ^a^	22.06 ^a^	4.06 ^b^	477.93 ^a^	43.00
Drying method means						
Parabola dome	41.14 ^c^	1.26 ^b^	14.29 ^b^	15.91 ^a^	427.58 ^c^	42.15 ^b^
Hot-air oven	50.24 ^b^	4.33 ^a^	21.98 ^a^	5.29 ^b^	438.06 ^b^	43.65 ^a^
Microwave	53.31 ^a^	4.48 ^a^	24.25 ^a^	4.42 ^c^	443.97 ^a^	43.35 ^a^
Significance of main effect and interaction						
BSFL types (A)	<0.001					0.769
Drying methods (B)	<0.001					0.002
A × B	<0.001					<0.001

^1^ ΔE**_Lab_* = √(
*L*_ref_ − L*_sample_*)^2^ + (*a*_ref_ − a*_sample_*)^2^ + (*b*_ref_ − b*_sample_*)^2^. ^a–e^ Means within columns with different superscript letters differ at *p* < 0.05. SEM = standard error of the mean. NDP = non-defatted BSFL dried with a parabola dome, NDH = non-defatted BSFL dried with a hot-air oven, NDM = non-defatted BSFL dried with a microwave, DP = defatted BSFL dried with a parabola dome, DH = defatted BSFL dried with a hot-air oven, DM = defatted BSFL dried with a microwave.

**Table 2 animals-13-02244-t002:** Particle size and distribution analysis of BSFL after different drying methods.

**Items**	**Sample (g)**													**Retain (%)**												
	**20** **(850 μm)**	**40** **(425 μm)**		**60** **(250 μm)**		**80** **(180 μm)**		**100** **(150 μm)**		**Pan**		**Total**		**20** **(850 μm)**		**40** **(425 μm)**		**60** **(250 μm)**		**80** **(180 μm)**		**100** **(150 μm)**		**Pan**		**Total**
Non-defatted BSFL																										
Parabola dome (NDP)	25.09 ^b^	71.01 ^a^		4.19 ^c^		0.00 ^c^		0.00 ^b^		0.00 ^b^		100.29		25.01 ^b^		70.80 ^a^		4.18 ^c^		0.00 ^c^		0.00 ^b^		0.00 ^b^		100.00
Hot-air oven (NDH)	21.53 ^b^	73.93 ^a^		4.44 ^c^		0.42 ^c^		0.00 ^b^		0.00 ^b^		100.32		21.47 ^b^		73.69 ^a^		4.43 ^c^		0.41 ^c^		0.00 ^b^		0.00 ^b^		100.00
Microwave (NDM)	42.15 ^a^	54.59 ^b^		3.58 ^c^		0.04 ^c^		0.00 ^b^		0.00 ^b^		100.36		42.00 ^a^		54.40 ^b^		3.56 ^c^		0.04 ^c^		0.00 ^b^		0.00 ^b^		100.00
Defatted BSFL																										
Parabola dome (DP)	4.21 ^c^	27.32 ^c^		45.55 ^a^		20.87 ^b^		2.05 ^b^		0.35 ^b^		100.34		4.20 ^c^		27.23 ^c^		45.39 ^a^		20.80 ^b^		2.04 ^b^		0.35 ^b^		100.00
Hot-air oven (DH)	4.32 ^c^	16.34 ^d^		40.99 ^ab^		26.62 ^a^		12.21 ^a^		0.11 ^b^		100.58		4.30 ^c^		16.24 ^d^		40.75 ^ab^		26.47 ^a^		12.14 ^a^		0.11 ^b^		100.00
Microwave (DM)	4.27 ^c^	15.58 ^d^		37.76 ^b^		25.12 ^a^		15.17 ^a^		2.34 ^a^		100.24		4.26 ^c^		15.55 ^d^		37.67 ^b^		25.06 ^a^		15.13 ^a^		2.33 ^a^		100.00
SEM	1.396	1.351		1.418		0.788		1.660		0.273				1.385		1.360		1.396		0.800		1.655		0.273		
BSFL types means																										
Non-defatted	29.59 ^a^	66.51 ^a^		4.07 ^b^		0.15 ^b^		0.00 ^b^		0.00 ^b^		100.32		23.49 ^a^		66.30 ^a^		4.06 ^b^		0.15 ^b^		0.00 ^b^		0.00 ^b^		100.00
Defatted	4.27 ^3^	19.75 ^b^		41.43 ^a^		24.20 ^a^		9.81 ^a^		0.98 ^a^		100.44		4.25 ^b^		19.67 ^b^		41.27 ^a^		24.11 ^a^		9.77 ^a^		0.93 ^a^		100.00
Drying methods means																										
Parabola dome	14.65 ^b^	49.16 ^a^		24.87 ^b^		10.44 ^b^		1.02 ^b^		0.17 ^b^		100.31		14.61 ^b^		49.01 ^a^		24.78 ^a^		10.40 ^b^		1.02 ^b^		0.17 ^b^		100.00
Hot-air oven	12.93 ^b^	45.13 ^b^		22.71 ^a^		13.52 ^a^		6.10 ^a^		0.06 ^b^		100.45		12.88 ^b^		44.97 _b_		22.59 ^ab^		13.44 ^a^		6.07 ^a^		0.06 ^b^		100.00
Microwave	23.21 ^a^	35.09 ^c^		20.67 ^c^		12.58 ^a^		7.59 ^a^		1.17 ^a^		100.31		23.12 ^a^		34.97 ^c^		20.61 ^a^		12.55 ^a^		7.57 ^a^		1.17 ^a^		100.00
Significance of main effect and interaction																										
BSFL types (A)	<0.001	<0.001		<0.001		<0.001		<0.001		<0.001				<0.001		<0.001		<0.001		<0.001		<0.001		<0.001		
Drying methods (B)	<0.001	<0.001		0.011		<0.001		<0.001		<0.001				<0.001		<0.001		0.010		<0.001		<0.001		<0.001		
A × B	<0.001	<0.001		0.028		0.002		<0.001		<0.001				<0.001		<0.001		0.026		0.003		<0.001		<0.001		
**Items**	**Cumulative (%)**												**Passing (%)**												***D_gw_* ^1^** **(μm)**	
	**20** **(850 μm)**		**40** **(425 μm)**		**60** **(250 μm)**		**80** **(180 μm)**		**100** **(150 μm)**		**Pan**		**20** **(850 μm)**		**40** **(425 μm)**		**60** **(250 μm)**		**80** **(180 μm)**		**100** **(150 μm)**		**Pan**			
Non-defatted BSFL																										
Parabola dome (NDP)	25.01		95.82		99.99		100.00		100.00		100.00		74.99		4.18		0.01		0.00		0.00		0.00		373.11 ^b^	
Hot-air oven (NDH)	21.47		95.16		99.59		100.00		100.00		100.00		78.53		4.84		0.41		0.00		0.00		0.00		363.67 ^c^	
Microwave (NDM)	42.00		96.40		99.96		100.00		100.00		100.00		58.00		3.60		0.04		0.00		0.00		0.00		414.98 ^a^	
Defatted BSFL																										
Parabola dome (DP)	4.20		31.43		76.82		97.61		99.65		100.00		95.80		68.57		23.18		2.39		0.35		0.00		233.03 ^d^	
Hot-air oven (DH)	4.30		20.54		61.29		87.75		99.89		100.00		95.70		79.46		38.71		12.25		0.11		0.00		208.52 ^e^	
Microwave (DM)	4.26		19.81		57.48		82.53		97.67		100.00		95.74		80.19		42.52		17.47		2.33		0.00		201.43 ^f^	
SEM																									1.102	
*p*-value																									<0.001	

^1^ Geometric mean diameter in μm by mass of sample. ^a–f^ Means within columns with difference superscript letters differ at *p* < 0.05. SEM: standard error of the mean. NDP = non-defatted BSFL dried with a parabola dome, NDH = non-defatted BSFL dried with a hot-air oven, NDM = non-defatted BSFL dried with a microwave, DP = defatted BSFL dried with a parabola dome, DH = defatted BSFL dried with a hot-air oven, DM = defatted BSFL dried with a microwave.

**Table 3 animals-13-02244-t003:** Nutritional value and chemical composition of BSFL after different drying methods.

Items	Nutrional Compositions (% of DM)									
	Moisture	Ash	CP	CF	EE	NFE	Ca	P	GE (kcal/kg)	pH
Non-defatted BSFL										
Parabola dome (NDP)	9.16 ^a^	5.08 ^f^	42.19 ^f^	7.32 ^e^	30.56 ^c^	5.69 ^e^	1.68	0.68 ^b^	5628.89 ^b^	6.79
Hot-air oven (NDH)	3.09 ^e^	5.76 ^d^	44.72 ^d^	7.20 ^e^	31.91 ^a^	7.32 ^d^	1.70	0.70 ^a^	5811.02 ^a^	6.82
Microwave (NDM)	3.45 ^d^	5.27 ^e^	43.16 ^e^	8.23 ^d^	30.90 ^b^	8.99 ^b^	1.71	0.70 ^a^	5861.21 ^a^	6.82
Defatted BSFL										
Parabola dome (DP)	4.77 ^b^	7.43 ^a^	57.18 ^b^	11.89 ^b^	12.94 ^d^	5.79 ^e^	1.68	0.70 ^a^	5035.33 ^d^	6.84
Hot-air oven (DH)	4.11 ^c^	6.49 ^b^	53.49 ^c^	14.82 ^a^	12.89 ^d^	8.20 ^c^	1.70	0.70 ^a^	5155.07 ^c^	6.80
Microwave (DM)	2.98 ^f^	6.07 ^c^	59.14 ^a^	9.84 ^c^	9.88 ^e^	12.09 ^a^	1.69	0.70 ^a^	5126.04 ^c^	6.82
SEM	0.019	0.023	0.178	0.036	0.082	0.029	0.007	NA	19.210	0.013
BSFL types means										
Non-defatted	5.23 ^a^	5.37 ^b^	43.36 ^b^	7.58 ^b^	31.12 ^a^	7.33 ^b^	1.70	0.70 ^a^	5767.04 ^a^	6.81
Defatted	3.95 ^b^	6.66 ^a^	56.60 ^a^	12.18 ^a^	11.90 ^b^	8.69 ^a^	1.69	0.69 ^b^	5105.48 ^b^	6.82
Drying methods means										
Parabola dome	6.97 ^a^	6.23 ^a^	49.68 ^b^	9.61 ^b^	21.75 ^b^	5.74 ^c^	1.68 ^b^	0.69 ^b^	5332.11 ^b^	6.80
Hot-air oven	3.60 ^b^	6.13 ^b^	51.15 ^a^	11.01 ^a^	22.40 ^a^	7.76 ^b^	1.70 ^a^	0.70 ^a^	5483.04 ^a^	6.83
Microwave	3.22 ^c^	5.67 ^c^	49.11 ^c^	9.03 ^c^	20.39 ^c^	10.54 ^a^	1.70 ^a^	0.70 ^a^	5498.62 ^a^	6.82
Significance of main effect and interaction										
BSFL types (A)	<0.001	<0.001	<0.001	<0.001	<0.001	<0.001	0.183	<0.001	<0.001	0.342
Drying methods (B)	<0.001	<0.001	<0.001	<0.001	<0.001	<0.001	0.006	<0.001	<0.001	0.115
A × B	<0.001	<0.001	<0.001	<0.001	<0.001	<0.001	0.178	<0.001	0.003	0.105

^a–f^ Means within columns with difference superscript letters differ at *p* < 0.05. SEM: standard error of mean. NA = not applicable. CP = crude protein, CF = crude fiber, EE = ether extract, NFE = nitrogen-free extract, Ca = calcium, P = phosphorus, GE = gross energy. NDP = non-defatted BSFL dried with a parabola dome, NDH = non-defatted BSFL dried with a hot-air oven, NDM = non-defatted BSFL dried with a microwave, DP = defatted BSFL dried with a parabola dome, DH = defatted BSFL dried with a hot-air oven, DM = defatted BSFL dried with a microwave.

## Data Availability

Not applicable.

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
