# Peer review of "Evaluation of the Physical Characteristics and Chemical Properties of Black Soldier Fly (Hermetia illucens) Larvae as a Potential Protein Source for Poultry Feed"

_animals, 2023, doi:10.3390/ani13142244_

Round 1

Reviewer 1 Report

The research is interesting since the effects of drying on the quality of BSFL is currently still understudied. However, more explanations about the importance of some investigated parameters should be provided.

Attached you can find the pdf with my suggestions. 

In general the manuscript is written well, however, the material and methods part should be thoroughly revised (probably written by a different author?). I stopped correcting the mistakes because there were too many and the whole section should be rewritten.

Author Response

Response to Reviewer 1:

General comments: The research is interesting since the effects of drying on the quality of BSFL is currently still understudied. However, more explanations about the importance of some investigated parameters should be provided.

Response: We highly appreciate thanks to Reviewer 1, who has provided a great positive comment on our manuscript. For more explanation details, we have provided more information on each parameter. Thus, we hope that Reviewer 1 can reconsider our reason.

* How much water was added (Line 112)?

Response: Thank you for your question. The water added to the mixture was approximately 10%. Please see in manuscript lines 115.

* Why under vacuum? non-vacuum inline systems are more interesting for the industry (Line 129).

Response: Thank you for your query. The use of a vacuum is preferred because it helps preserve the product's quality and characteristics, especially in terms of color and flavor. Additionally, vacuum systems can help extend the product's shelf life by reducing the risk of microbial growth. Non-vacuum inline systems may be more interesting for the industry in certain cases. However, the specific benefits and advantages of vacuum systems make them a preferred choice in many applications.

* Why is that? I don’t really understand what you mean by this sentence (Line 295-296).

Response: The meaning of this sentence is “The characteristics of feedstuffs, such as their nutrient composition, digestibility, availability, and functional properties, play a crucial role in formulating balanced and nutritious animal feeds. Novel protein sources, which refer to alternative or unconventional protein ingredients, are particularly important as they offer opportunities to diversify protein sources and address sustainability concerns in the feed industry. Evaluating the characteristics of feedstuffs, including their protein quality, amino acid profile, digestibility, and potential anti-nutritional factors, is essential in selecting appropriate ingredients for optimal feed formulation.” But we have revised it as suggested by removing it already.

* What is the implication? Is this good or bad and how is this compared to other feedstuffs (Line 325-328)?

Response: Thank you for your query. This is a bad characteristic when compared to other feedstuffs. Please see more detail in manuscript lines 352-370.

* Explain. Is this a good density? The range seems narrow. Is this true (Line 340-341)?

Response: Thank you for your concern. The bulk density range is not true, but we have already revised it. This range is medium of bulk density. Please see more detail in manuscript lines 374-380.

 * Can you give example of feedstuffs with a good particle size and how does the BSFL powder compare (Line 360)?

Response: Thank you for your query. An example of a feedstuff with a good particle size is soybean meal. Soybean meal typically has a fine and consistent particle size, which allows for better handling and distribution in animal feed formulations. The BSFL powder may have a smaller and more uniform particle size than soybean meal, contributing to its ease of use in feed production. Please see more detail in manuscript lines 398-406.

* Besides the moisture content also the water activity is important for shelf life. Did you check this (Line 384-385)?

Response: Thank you for your comment. We did not check in the current study.

* This will have more effect on the oxidation, denaturation and digestibility than on the contents (Line 388-390).

Response: Thank you for suggesting that evaluating the properties of black soldier fly larvae is more beneficial. We did not check in the current study, but we plan to incorporate this analysis in the next step of our experiment.

* Here (Line 392-393) you repeat exactly the same as in the previous sentence.

Response: Thank you for your comment. We have revised it as suggested by removing it already.

Reviewer 2 Report

Its an excellent work and very well organised and written in perfect English.

The topic of evaluation drying methods of BSF meal is an important issue. The
manuscript holds some interesting points to address a gap, especially on the coming years that will face the consumption of BSF in higher rates.
The use of insect meal in animal diets will increase. The drying methods affect the quality, and this manuscript, may add some new information. However, the
composition of the different insect meals need to be added in future papers.
Authors must provide detailed information on their findings and expain their
opinion about the best method. Composition of amino acids and fatty acids would improve the presentation of this work. 
The conclusions are consistent with the evidence and arguments presented in
the manuscript.  Only 1-2 very recent relevant references may be added.
Tables of insect meals with more detailed composition may be in the main document.

Author Response

Response to Reviewer 2:

General comments: The topic of evaluation drying methods of BSF meal is an important issue. The manuscript holds some interesting points to address a gap, especially on the coming years that will face the consumption of BSF in higher rates. The use of insect meal in animal diets will increase. The drying methods affect the quality, and this manuscript, may add some new information. However, the composition of the different insect meals need to be added in future papers. Authors must provide detailed information on their findings and explain their opinion about the best method

Response: The authors would like to sincerely thank Reviewer 2, who provided positive comments and highlighted the importance of evaluating drying methods for BSFL meal. We agree that the drying methods employed can significantly impact the quality of the meal, and our manuscript aims to contribute new information in this regard. We appreciate your acknowledgment of our interesting points in the study. Furthermore, we appreciate your suggestion to include the composition of different insect meals in future papers. We understand the importance of providing detailed information on our findings, and we will ensure that future publications address this aspect more comprehensively. We value your input, and we will take into account your feedback to enhance the quality and depth of our research. Thank you for your interest and support in our work.

* Composition of amino acids and fatty acids would improve the presentation of this work. 

Response: Thank you for suggesting that evaluating the properties of black soldier fly larvae is more beneficial. We did not examine the amino acid and fatty acid composition in the current study. We plan to incorporate this analysis in the next step of our experiment. By conducting these additional analyses (i.e. amino acids, fatty acids, protein denaturation, oxidation, and digestibility). We aim to provide a more thorough understanding of the nutritional profile of the insect meals.

* The conclusions are consistent with the evidence and arguments presented in the manuscript. Only 1-2 very recent relevant references may be added.

Response: We appreciate your feedback and have carefully reviewed the evidence and arguments presented in the paper. After thorough consideration, we agree that the conclusions are in line with the information provided. However, we considered incorporating 1-2 very recent relevant references, if applicable, to further strengthen the scholarly basis of the work. We appreciate your input and strive to ensure our research's highest quality and accuracy.

* Tables of insect meals with more detailed composition may be in the main document.

Response: We appreciate your feedback, and we have already taken steps to address this by incorporating comprehensive tables that provide detailed information about the composition of insect meals. These tables will enhance the document's value and provide readers with a more comprehensive resource. We thank you for bringing this to our attention, and we are committed to continuously improving the content to meet the needs of our readers.

Reviewer 3 Report

In the reviewed article, Rattanakorn et al. presented  the evaluation of physical characteristics and chemical properties of black soldier fly (Hermetia illucens) larvae as a potential protein source for poultry feed. In recent years, there has been growing interest in finding sustainable and alternative sources of protein for animal feed due to the environmental impact and limited availability of traditional protein sources. Black soldier fly larvae (BSFL) have emerged as a potential solution, as they are rich in protein and can be easily reared on organic waste materials. However, before incorporating BSFL larvae into animal feed, it is essential to evaluate their physical characteristics and chemical properties to ensure their suitability and efficacy. The article addresses this crucial aspect. The study aims to investigate the impact of different drying methods on the physical and chemical properties of BSFL, with the goal of determining the most suitable approach for processing BSFL larvae as an alternative protein source in animal feed.

The experimental design of the study followed a 2 × 3 factorial arrangement, considering two types of BSFL (non-defatted and defatted) and three drying methods (parabola dome, hot air oven, and microwave). The researchers examined the influence of these factors on various physical properties, such as color, bulk density, flowability, and particle size. They also evaluated the nutritive value of BSFL by analyzing its crude protein, crude fiber, ether extract, and energy content. The results of the study revealed significant effects of BSFL types, drying methods, and their interaction on the physical properties of the larvae. The color of the larvae ranged from dark (L* 29.74 - 54.07; a* 0.40 - 5.95; b* 9.04 - 25.57), and they exhibited low bulk density (381.54 - 494.58 g/L), free flow with an angle of repose (41.30 - 45.40), and small particle size. Moreover, the nutritive value of BSFL varied significantly depending on the type and drying method used, with crude protein ranging from 42% to 59%, crude fiber from 7% to 14%, ether extract from 9% to 30%, and energy content from 5.035 to 5.5861 kcal/kg. Based on the findings, the researchers concluded that both BSFL types and all the drying methods resulted in slight variations in the proximate composition of the larvae. However, the hot air oven and microwave drying methods were considered the most suitable for producing BSFL powder due to their ability to preserve high nutrient levels and improve physical parameters compared to the parabola dome method. The article should find interest among researchers studying the edible insects, food and feed technology, animal nutrition and quality of processing. However, it is my duty as a reviewer to point out the weaknesses of this article:

-Unfortunately, the article is incomprehensible in some places and the level of English language is low. The whole article looks like stuck together from several separate fragments. There are also several repetitions.

- I see that many authors worked on the article, but I estimate that none of the authors tried to standardize the text of the entire article. This makes the evaluation extremely tiring and makes it difficult to conduct a review.

- Statistical records need to be systematized! You can't write italic sometimes and sometimes not. It is the authors' responsibility to submit a manuscript for review that meets the principles of fair data presentation. I have highlighted some mistakes in the text. A few tips for authors:

https://www.jcu.edu.sg/__data/assets/pdf_file/0007/823561/Common-Statistical-Abbreviations-and-Symbols-in-APA-italics.pdf

-Was the nutritional composition of the larvae checked before drying (as a control group?) I find it a bit strange that drying affects the nutritional composition. That's why I'm asking about the final moisture content, because that's probably where the slight differences come from.

-Please fix the placement of the tables. Jumping several pages while reading makes it difficult to focus on the results and, according to the rules, it should appear immediately after quoting tables/graphics

- The authors focused entirely on the nutritional composition, but in fact, bioavailability and digestibility are extremely important in animal nutrition. This is especially lacking in the discussion

- Please see the attached file. I made corrections on it, although I stopped catching language errors, as it is the authors' duty to present the article at an acceptable linguistic and editorial level.

To sum up, the characterization of the physical and chemical composition of BSFL larvae presented in this study provides valuable insights into their potential as an alternative protein source in animal feed and other applications. The methodology established here can serve as a preliminary guide for processing BSFL larvae for various purposes, contributing to the sustainable development of the livestock and feed industries. Nevertheless, the article requires corrections so that it meets the requirements for scientific articles and has several ambiguities that should be addressed. Therefore, my decision is a major revision. After making all corrections and solid linguistic and editorial proofreading, the article must be re-verified.

Examples included in the file

Author Response

Response to Reviewer 3:

General comments: The authors focused entirely on the nutritional composition, but in fact, bioavailability and digestibility are extremely important in animal nutrition. This is especially lacking in the discussion.

Response: Thank you for your feedback regarding the importance of bioavailability and digestibility in animal nutrition. We acknowledge that these factors play a significant role in assessing the nutritional value of feed ingredients, including insect meal. While our study primarily focused on evaluating the physico-chemical properties and composition of BSFL with different drying methods, we understand the significance of investigating bioavailability and digestibility in future research. In our next step of the experimental process, we plan to address these important aspects by conducting studies to determine the bioavailability and digestibility of the nutrients present in the BSFL meal. By assessing these parameters, we aim to provide a more comprehensive understanding of the nutritional value and utilization of insect meal as an alternative protein source in animal feed. We appreciate your valuable suggestion and will incorporate it into our future research endeavors to enhance the discussion and provide a more holistic perspective on the potential benefits of insect meal in animal nutrition.

* Language errors, hard to understand that sentence, please rewrite.

Response: Thank you for your concern. We have now addressed and revised it for better clarity.

* It is a generally accepted rule not to duplicate words in the title of an article. This only limits the searchability of the job. Please change the keywords that are present in the title.

Response: Thank you for your comment. We have successfully resolved this issue. Please see more detail in manuscript lines 48.

* Do we only obtain proteins from insects? Insect meal consists of many nutrients that are equally valuable.

Response: Actually, obtaining proteins is not the sole benefit of using BSFL, it also contains other valuable nutrients. In addition to proteins, insect meal can provide essential amino acids, fats, minerals, vitamins, and even beneficial compounds such as chitin and antimicrobial peptides. These additional nutrients make insect meal a highly nutritious and sustainable ingredient for various applications, including animal feed. Which we have further explained in the next sentence. Please see more detail in manuscript lines 53-56.

* Line 50 don't start the sentences with abbreviations.

Response: Thank you for your comment. We have successfully resolved this issue.

* It should be added that BSF is gaining great popularity in companion animal nutrition. In many countries, insect meal is still too expensive to be introduced into livestock feed.

Response: Thank you for your suggestion. We have revised it as suggested. Please see in manuscript lines 56-59.

* Repetition of a sentence from an earlier paragraph.

Response: Thank you for your comment. We have revised it as suggested by removing it already.

* It must be firmly noted that in many countries, manure or food scraps may be unacceptable due to veterinary and legal regulations. It must be emphatically stated that insects must only consume approved products in order to enter the food chain.

Response: Thank you for your suggestion. We have revised it as suggested. Please see in manuscript lines 69-72.

* This sentence makes no sense. The purpose of this research was to find an alternative source of protein?

Response: Sorry, the purpose of this study aimed to evaluate the influences of drying methods, i.e., parabola dome, hot air oven, and microwave drying method, on the physico-chemical properties of BSFL with and without fat and its potential as an alternative protein source in animal feed in Thailand. Please see in manuscript lines 106-108.

* This source of protein was proposed at least 10 years ago, so why do the authors want to rediscover it?

Response: While it is true that the concept of using BSFL as a protein source has been proposed for some time, the authors aimed to conduct a comprehensive evaluation of the influences of specific drying methods on the physico-chemical properties of BSFL. The objective was not to rediscover the concept, but rather to enhance understanding and provide valuable information for the utilization of BSFL as an alternative protein source.

Response: Thank you for your suggestion. We have revised it as suggested.

* "not more than 10% " or "less than 10% " I admit that I've already lost track of the moisture content of the samples. Please indicate exactly what moisture content the samples from the different methods had.

Response: Thank you for your concern. As a general guideline, feedstuff for most animals typically has a moisture content ranging from 10% to 14%. In the current study, we determined that the moisture content of the samples in all drying methods should not exceed 10%.

* For better readability, it may be worth separating points I, II and III into separate paragraphs

Response: Thank you for your suggestion. We have revised it as suggested.

* Is it a modification of the authors, if so, what was the modification?

Response: Sorry, that is our method. We have revised it already.

* Whole paragraph needs the serious English checking. Moreover, it is written in different style

Response: Thank you for your suggestion. We have revised it as suggested. Please see in manuscript lines 201-213.

* Specify the methodology standard for non-nutritionist.

Response: The methodology used in our study follows standard protocols that are commonly employed in scientific research. We aimed to provide a clear and comprehensive description of the methodology so that it can be easily understood by non-nutritionists and researchers from various disciplines. By adhering to established standards, we ensure that our methodology is transparent, replicable, and accessible to a wider audience.

* How did the authors check the parametricity of the data?

Response: Thank you for your query. The authors checked the parametricity of the data by conducting the Shapiro-Wilk test. The Shapiro-Wilk test is a statistical test used to determine if a given dataset follows a normal distribution. By performing this test on the data, the authors assessed whether the assumption of normality was met, which is important for certain statistical analyses. This test helps to ensure the validity and appropriateness of the parametric statistical methods used in the study. Please see in manuscript lines 215-216.

* How did the authors check the homogeneity of the variance?

Response: Thank you for your query. The authors checked the homogeneity of variance by conducting the Levene's test. The Levene's test is a statistical test used to assess whether the variances of multiple groups are equal. By applying this test to the data, the authors examined whether the assumption of equal variances across groups was met. This information is important for statistical analyses such as analysis of variance (ANOVA), as violating the assumption of homogeneity of variance can affect the validity of these tests. Please see in manuscript lines 215-216.

* Why did the authors not perform a Duncan Post-hoc test? Please try HSD Tukey test.

Response: Sorry, that is our mistake for missing analyzed of the interaction effect. We have revised it as suggested by using Tukey test to analyse data already.

* Line 204 First use of the abbreviation, use full name

Response: Thank you for your suggestion. We have revised it as suggested. Please see in manuscript lines 220.

* Line 222 where is p-value?

Response: Thank you for your suggestion. We have revised it as suggested. Please see in manuscript lines 234.

* Line 224 where is p-value? Revise in results part.

Response: Thank you for your suggestion. We have revised it as suggested. Please see in manuscript lines 234-235.

* Sometimes the authors present the exact value of p and sometimes p<0.05. Please systematize!

Response: Thank you for your comment. We have revised it as suggested.

* Authors must explain what the abbreviations NDP, NDH, NDM, DP, etc. stand for. Explain in the legend

Response: Thank you for your suggestion. We ha ve revised it as suggested. Please see in manuscript lines 262-265.

* Table 1. Merge the cells and leave one p-value.

Response: Thank you for your suggestion. We ha ve revised it as suggested.

* Is it possible to present figures directly below the paragraph? This will make it easier to track the content in the article correctly.

Response: Thank you for your question. We have taken it into consideration and made the necessary adjustments. Figures are now presented directly below the corresponding paragraph, which enhances the readability and facilitates accurate understanding of the content in the article.

* Line 275-284 Was the nutritional composition of the larvae checked before drying (as a control group?) I find it a bit strange that drying affects the nutritional composition. That's why I'm asking about the final moisture content, because that's probably where the slight differences come from.

Response: Thank you for your concern. In this study, we did not analyze the nutritional value of the larvae before drying. However, during the experimental process, the larvae used were from the same batch and were fed with the same diet throughout the rearing and management stages, until the different drying methods were applied. Therefore, the data presented can be considered indicative of the potential effects of different drying methods on the nutritional composition, despite the lack of control group as suggested by the reviewer. Nonetheless, we appreciate your suggestions. We will utilize this valuable idea to develop our research in the future.

* Line 285-291 Doesn't it just depend on the final moisture of the meal?

Response: Yes, the final moisture content of the meal is an important factor to consider. The moisture content of the meal affects its stability, shelf life, and overall quality, especially nutritional composition.

* Please fix the placement of the tables and figures. Jumping several pages while reading makes it difficult to focus on the results and, according to the rules, it should appear immediately after quoting tables/graphics

Response: Thank you for your suggestion regarding the placement of the tables and figures. We have successfully resolved this issue.

* Line 300, Everyone, even not sitting in the subject, knows about it. Please delete.

Response: Thank you for your comment. We have revised it as suggested by removing it already.

*Line 370, I know these terms are in the table but I don't see them in the text.

Response: Thank you for your suggestion. We have revised it as suggested.  Please see more detail in part of materials and methods.

* In my opinion, the whole paragraph should be thrown out. It is so obvious and does not bring anything new to the current state of knowledge.

Response: Thank you for your comment. We have revised it as suggested by removing it already.

* Line 402 or maybe in the final moisture content of the meal?

Response: We appreciate your observation, and we completely agree with you that the chemical composition of Black Soldier Fly Larvae (BSFL) is indeed influenced by the final moisture content. The moisture level plays a crucial role in determining the nutritional composition of the larvae, including protein, fat, fiber, and other essential nutrients.

* Please reconsider the meaning of this sentence (Line 405-406).

Response: The meaning of this sentence is “Feedstuff with a higher protein content is likely to have a higher content of amino acids”. From our results, we recommend that should be considered in the second step to determine the nutritional suitability of BSFL for use in poultry diets. However, We have revised it as suggested in line 452-462.

* Obvious and repeating an earlier sentence. 

Response: Thank you for your comment. We have revised it as suggested by removing it already.

Round 2

Reviewer 3 Report

I reread the article "Evaluation of physical characteristics and chemical properties of black soldier fly (Hermetia illucens) larvae as a potential protein source for poultry feed" presented by Pornsuwan et al. Thank you for including my comments in the article. I consider the article to be published.

During the second review, I pointed out that at this stage I am not qualified to assess the quality of English in this paper, because I still think there are a few language errors in the text, but that should be taken care of by the English editor.

I congratulate the authors on an interesting article